# The Relationship between Workaholism and Personal Burnout in Dual-Earner Couples: An Analysis Using the Actor-Partner Interdependence Model

Eleonora Russo [1], Pawel Atroszko [2], Sara Zaniboni [1,3], Stefano Toderi [1] and Cristian Balducci [4,*]

1 Department of Psychology, University of Bologna, 40127 Bologna, Italy; eleonora.russo10@studio.unibo.it (E.R.); sara.zaniboni4@unibo.it (S.Z.); stefano.toderi@unibo.it (S.T.)
2 Department of Psychology, University of Gdansk, 80-309 Gdansk, Poland; p.atroszko@ug.edu.pl
3 Department of Management, Technology and Economics, ETH Zürich, 8092 Zürich, Switzerland
4 Department for Life Quality Studies, University of Bologna, 47921 Rimini, Italy
* Correspondence: cristian.balducci3@unibo.it

**Abstract:** This study tested the workaholism–personal burnout relationship by using the Actor-Partner Interdependence Model in a sample of 138 Italian dual-career couples. Specifically, in line with previous literature, the presence of actor and partner 'effects' was hypothesized, whereby the level of workaholism of men and women influences their own level of personal burnout (actor effect) and that of their partners (partner effect). In addition, the moderating role of the presence and number of children in the relationship between workaholism and personal burnout was also analyzed. The results confirmed a significant actor effect. In contrast, the partner effect was partially confirmed (only for the pathway from female partner workaholism to male partner personal burnout). In addition, the presence of children played a key role. First, it strengthened the positive relationship between the female partner's workaholism and the male partner's personal burnout. Second, it attenuated the positive relationship between a female's workaholism and her own personal burnout. The obtained results are discussed, and based on these, considerations are provided on possible interventions to decrease the potential impact of workaholism on couples' well-being and the implications for a sustainable work and family life.

**Keywords:** workaholism; burnout; actor-partner; dual-career couples; children; work-family balance; gender

## 1. Introduction

In modern work environments, increasingly characterized by high competition both between and within organizations and the digitalization and intensification of work, workers are frequently exposed to a pressing request for a heavy work investment (i.e., time and effort investments), which often leads to very high levels of job stress and to a concrete threat to their mental well-being [1–4]. Such trends and their potential consequences might have been even accentuated by the recent pandemic [5]. One of the by-products of the above work environmental conditions may be workaholism, which has been initially defined as "the compulsive and uncontrollable need to work incessantly" [6]. Indeed, although the genesis of the phenomenon is still unclear and may be related to certain underlying dispositional characteristics such as obsessive-compulsive personality [7,8], research has shown that working conditions such as a chronically high workload may be implicated [9]. This suggests that workaholism may be a dysfunctional coping pattern developed, at least in part, in response to a mix of distressing job-related factors.

Right from the start, research has highlighted two basic elements of workaholism: working excessively, and an irresistible inner urge to work [10], leading to the idea that

workaholism regards 'compulsive overworking' and that it can be viewed as a true behavioral addiction [11–17]. Workaholics are individuals who work too much and spend little time with their families, driven by a high dysfunctional intrinsic motivation and lack of control over work. At the same time, they experience little pleasure while working (Balducci et al., 2021) [18]. In addition, workaholism is linked to a variety of negative health-related conditions, including among others burnout, anxiety, depression, bipolar and personality disorders [10,19], sleep problems [20], and increased cardiovascular risk [21], making the consequences of the phenomenon particularly serious from an individual perspective. The organizational implications of workaholism should not be underestimated too [22], given that the tendency is often shown by managers [23] who may pass it via role modeling to their subordinates or by creating stressful workaholic environments. Additionally, workaholism is linked to presenteeism [24,25] and fuels less prosocial [26] and more aggressive behaviors [27] at work, while at the same time having null or negative implications for a positive job performance [18].

While most research on workaholism to date has focused mainly on refining definitional aspects of the construct and documenting its individual correlates and consequences [10,16,28], relatively less is known about the implications of the phenomenon in the family and particularly whether and how it influences family dynamics and functioning. The initial work on this carried out by Robinson [13], clearly highlighting the costs of workaholism for close others, was not followed up by the deserved attention. Therefore, building on spillover-crossover theory [29] and on models of mental disorders contagion [30], and using an actor-partner interdependence approach [31,32], we focus here on dual-earner couples and investigate the potential reciprocal influences of each partner's levels of workaholism on the other partner's personal burnout, that is, the degree of physical and psychological fatigue experienced by the person [33]. Additionally, we explore whether the presence of children in the family, which increases the salience of the family role at the expense of the work role—a circumstance that may be particularly troubling for individuals with workaholic tendencies—may amplify the relationships between workaholism and personal burnout. In the following paragraphs, we develop theoretical arguments and review previous empirical findings as a basis for the formulation of the study hypotheses.

*Study Theoretical Models and Hypotheses*

Spillover-crossover theory has been proposed to explain the ways in which feelings and experiences are carried over from the work domain to the family domain. The concept of "spillover" is generally used to explain work–family conflict processes, whereby the boundaries between the two domains become permeable to the point that one domain slips over the other [34]. This can occur either from the work domain to the family domain or vice versa. Therefore, spillover can be defined as a within-person, across-domains transmission of demands and consequent strain from one domain to another domain [35]. Differently, the notion of "crossover" describes a dynamic in which the feelings and affective states felt in one role by an individual influence the emotions of the partner in the same role [36,37]. In other words, crossover involves transmission across individuals. To better understand the dynamics by which workaholism might affect the quality of a couple's life, Bakker et al. [36] proposed the spillover-crossover model. This model emphasizes the presence of a dual process, in which work/family-related tension first spills over to the work/family sphere (spillover) and then is transmitted to others (crossover).

In a study of dual-earner couples [36] it was shown that workaholic partners are less involved in family duties, thus creating more interpersonal conflict, and providing less support to their partner (spillover), causing a decrease in couple satisfaction (crossover). Another study by Shimazu et al. (2011) [38] found that a female partner's workaholism level had a positive influence on the male partner's perception of work-family conflict, supporting a unidirectional crossover dynamic. In a more recent study, Clark et al. (2021) [39] documented that on days in which the study participant reported more workaholic cog-

nition and behavior, their partner reported more stress and relationship tension in the evening. In other words, Clark et al. (2021) [39] supported a direct crossover of workaholism impact, whereby workaholics' feelings and tensions are transmitted directly to their partners, negatively affecting their well-being [39]. These results are plausible since workaholism is associated with negative withdrawal feelings when the individual is not at work, such as anger and irritability, anxiety, and guilt [16,17]. These feelings may be absorbed directly by the other member of the couple (i.e., without postulating further intervening variables such as work-family conflict), leading to a lower level of well-being in the partner. Such transmission mechanisms may be explained by direct contagion processes, which have been well documented in the literature [30]. Specifically, in the case of emotional contagion, stress-related emotions can be easily identified in facial expressions and postures [40,41], with the observer showing an automatic tendency to synchronize with the expressions displayed by others, leading the individual to experience the same emotions as the counterpart [42,43].

However, Clark et al.'s (2021) [39] study focused on the workaholism level of only a member of the couple, ignoring reciprocal stressor-strain crossover dynamics. Additionally, previous research did not test for potential moderators of these dynamics, which may help to reach a more fine-grained view of boundary conditions under which they operate within couples. Thus, based on the above considerations and available findings, in the present study, we investigate crossover relationships involving workaholism and personal burnout in dual-earner couples. Additionally, we explore the moderating role of the presence of children in these dynamics. To do this, we make use of the actor Actor-Partner Interdependence Model (APIM) [31,44], which is particularly useful for assessing the level of interdependence and mutual influence of different types of dyads, in which one person's feelings, cognitions, and behaviors influence the feelings, cognitions, and behaviors of the other. In its basic form, the APIM includes two independent variables and two dependent variables, the study of which can prove useful to understand the presence of 'actor effects' (i.e., how much a person's level of workaholism is related to their own levels of personal burnout) and 'partner effects' (how much a person's level of personal burnout is related to their partner's level of workaholism). A graphical representation of the conceptual model adopted is reported in Figure 1.

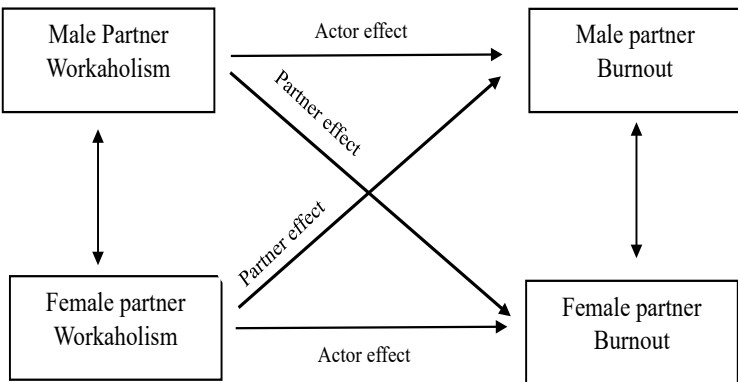

**Figure 1.** The actor-partner interdependence model of workaholism.

First, we will test for the actor effects, specifically hypothesizing that for both male and female partners, a higher level of workaholism will be related to one's own level of personal burnout (Hypothesis 1). This is, at least partially, a replication hypothesis, since the relationship between workaholism and burnout symptoms is well-established [10]. However, we focus here on personal burnout, which may not be necessarily related to work. Additionally, the adoption of the APIM model will allow us to assess whether the relationship holds at the same time for both partners of the couple and whether the relationship is different, in magnitude, in the two genders.

Second, we will test for partner effects and, building on the idea of direct crossover as explained by contagion processes (see above), we hypothesize that the level of workaholism reported by one partner will be positively related to the level of personal burnout of the other partner (Hypothesis 2). With the exception of the preliminary evidence provided by Clark et al. (2021) [39], previous studies have failed to support a direct impact/relationship between one partner's level of workaholism and the other partner's level of stress and unwell-being [38], meaning that there is a need of more work on this. Additionally, testing the formulated hypotheses with the APIM model will also allow us to observe whether, in dual-earner couples, the levels of workaholism of both partners tend to converge, that is, are positively related, which would be additional evidence confirming the existence of crossover processes operating at different levels in the couple.

Third, we will explore the role that the presence of children has in the different workaholism–personal burnout relationships implied by the adopted model. Although there is some evidence that workaholic tendencies in parents are related to their children's emotional and behavioral problems [45], little is known about whether the presence (and the number) of children in the family has health and well-being implications for parents with workaholic tendencies. Based on role theory [46], children may amplify role problems for individuals with workaholic tendencies, since children require presence and care, thus making more salient and important the role of caregiver at the expense of that of worker, limiting in workaholics the desired level of investment in work-related activities. This should lead to higher levels of distress and personal burnout. Additionally, the presence of children and thus the need to invest in the family may accentuate the withdrawal symptoms of workaholics when they are not at work, increasing negative contagion (crossover) processes, and leading to higher levels of partner's personal burnout. Thus, we hypothesize that both the actor and partner effects relating workaholism to personal burnout will be accentuated, in dual-earner couples, by the number of children in the family (Hypotheses 3). Although this may be true in general, implying that the presence of children would strengthen all the workaholism–personal burnout relationships examined, it is also true that traditional gender role expectations [47,48] suggest that presence and care for children are mainly the responsibility of women. This means that female partners would perceive a higher pressure to devote time to the family in the presence of children, which may generate stronger role (conflict) problems if they have workaholic tendencies. Thus, in testing for our third hypothesis we will also look at whether, with the presence of children in the family, it would be particularly the actor and partner 'effects' initiated by female partner workaholism (see Figure 1) that will be accentuated in terms of their personal burnout implications.

## 2. Materials and Methods

### 2.1. Participants

The sample consisted of 276 participants distributed in 138 couples. They were contacted and invited to take part in a questionnaire-based study on work-related health and well-being. The participants' ages ranged from 24 to 71 years for men (M = 48.8; median = 51; SD = 10.75), and from 26 to 66 years for women (M = 45.6; median = 46; SD = 10.03). All participants were from Italy. Most of the male and female participants had a high school diploma (41.3% of men, 48.6% of women). Further, a high percentage of participants had a bachelor's degree (32.6% of men and 42% of women), while the remainder held a middle school diploma (8.7% of women, compared to 26.1% of men). Most of the men worked as freelancers/self-employed (53.6%), such as for example engineers, designers, and condominium administrators, while employees made the remaining part. In contrast, there was a prevalence of women who worked as employees (59.5%). Most couples had children (M = 1.54; SD = 0.65). The general trend was having two children (38.4%).

### 2.2. Procedure

We adopted a convenience sampling method. First, with the help of master's students enrolled in our courses in Italy, we contacted people by phone or e-mail in the researchers' network of acquaintances. We looked for couples where both members were full-time workers and in which at least one member of the couple had a responsibility position at work, which is usually a risk factor for reporting higher levels of workaholism [49]. This strategy aimed to have the construct focused by the study (i.e., workaholism) well-represented in the sample. The study's objectives were explained, and the potential participant was asked to invite his/her partner to participate in the study. We obtained a refusal in approximately 8% of the total number of workers contacted. Those who agreed to participate were asked to explain the aim of the study to their partner and to provide their own and their partner's e-mail address. The questionnaire was delivered via the web by using the platform Qualtrics, which complies with the EU General Data Protection Regulation (GDPR). The questionnaire link was sent to the email addresses received by the participants. Data of the two members of each couple were linked by using the same anonymous code generated by the participants based on factual personal information.

### 2.3. Ethical Aspects

The study did not involve medical treatment or other procedures that could cause psychological or social discomfort to participants, who were all adult healthy subjects. Thus, ethical approval was not requested. However, the study was conducted in line with the Helsinki Declaration, as well as the data protection regulation of Italy. Participation in the research was voluntary and not rewarded; data were treated to preserve the confidentiality of responses, and the analyses were conducted on anonymous data. The cover page of the questionnaire provided information about the study aims and explained that participation in the study was voluntary and could be stopped at any stage. The cover page also explained how the data would be treated and provided instructions on how to fill out the questionnaire. It was also emphasized that going on to fill out the questionnaire corresponded to giving informed consent for participation. Finally, the cover page provided an e-mail address that could be used to ask for additional information regarding the study, including feedback about the provided responses.

### 2.4. Measures

Participants were first asked to supply sociodemographic data. After this, we measured the main study constructs using the following tools.

#### 2.4.1. Workaholism

To assess participants' level of workaholism, the "Multidimensional Workaholism Scale- MWS" [17] was used. Items were adapted into Italian by using the back translation method. The MWS is based on 16 items that assess four different aspects of workaholism. These aspects correspond to four subscales: Motivational (e.g., "I always feel an inner tension that drives me to work"); Cognitive (e.g., "At any given time, most of my thoughts are directed toward work"); Emotional (e.g., "I am almost always frustrated when I cannot work"); and Behavioral (e.g., "I tend to work more hours than most of my colleagues"). Each subscale includes four items. Participants indicate the frequency of experience for each aspect investigated by the scale items on a 5-point Likert scale (1 = never; 5 = always). The final score is based on the average of the individual items, with a high score indicating a high risk of workaholism. An overall workaholism score can be computed [17]. The MWS has shown significant incremental validity in comparison to well-established measures of workaholism, namely "Dutch Work Addiction Scale-DUWAS" [14], "Work Addiction Risk Test-WART" [13], and "Workaholism Battery-WorkBAT" [15] in predicting emotional exhaustion, negative work-related rumination, and depressive symptoms. The Italian version of the scale has been validated [50]. The scale properties obtained in the present study, are reported in Table 1.

**Table 1.** Mean, SD, Cronbach α, and correlations of the main study variables.

| | M (SD) | α | 1 | 2 | 3 | 4 | 5 | 6 | 7 | 8 |
|---|---|---|---|---|---|---|---|---|---|---|
| 1 Male partner WL | 2.83 (0.76) | 0.910 | 1 | | | | | | | |
| 2 Male partner Bout | 2.61 (0.71) | 0.850 | 0.60 *** | 1 | | | | | | |
| 3 Female partner WL | 2.87 (0.73) | 0.902 | 0.17 P = 0.05 | 0.24 ** | 1 | | | | | |
| 4 Female partner Bout | 3.04 (0.71) | 0.833 | 0.09 | 0.23 ** | 0.45 *** | 1 | | | | |
| 5 N. of children | 1.53 (1.15) | | 0.17 * | 0.08 | 0.03 | 0.12 | 1 | | | |
| 6 Male partner job p. (1 = high) | | | 0.34 *** | 0.20 ** | −0.12 | −0.09 | 0.21 * | 1 | | |
| 7 Female partner job p. (1 = high) | | | −0.02 | −0.01 | 0.15 | −0.16 | −0.11 | −0.07 | 1 | |
| 8 Male partner educ. (1 = univ.) | | | −0.07 | −0.05 | 0.05 | −0.02 | −0.22 ** | −0.06 | −0.15 | 1 |
| 9 Female partner educ. (1 = univ.) | | | −0.07 | −0.14 | −0.02 | 0.01 | −0.29 *** | −0.29 *** | 0.02 | 0.30 *** |

Notes: Male partner WL = Male partner workaholism; Male partner Bout = Male partner personal burnout; Female partner WL = female partner workaholism; Female partner Bout = female partner personal burnout; N. of children = number of children; Male partner job p. = Male partner job position; Female partner job p. = female partner job position; High = managerial level and self-employed (vs. Low = all others); Male partner educ. = Male partner educational level; Female partner educ. = female partner educational level; Univ. = university level (vs. Low = up to high school diploma). *** $p < 0.001$; ** $p < 0.01$; * $p < 0.05$.

### 2.4.2. Personal burnout

To measure the participants' level of personal burnout, the "Copenhagen Burnout Inventory-CBI" [36] was adopted. The scale was initially tested on a sample of human service workers. The CBI is a self-reported measure comprising 19 items. The items are based on three main sub-scales, representing three different dimensions of burnout: personal burnout (six items, e.g., "How often do you feel tired?"); work-related burnout (seven items, e.g., "Are you exhausted in the morning at the thought of another day's work?"); and client-related burnout (six items, e.g., "Do you find it difficult to work with clients?"). A high score means high burnout symptoms. Burnout related to only one dimension can also be calculated by summing only the score of the specific subscale. The three subscales have high internal reliability and are useful for differentiating jobs [33]. The Italian version of the scale was used for this study [51]. The scale was tested on a sample of Italian teachers of different grades in the 2010/2011 school year for the Italian validation. The six items making up the personal burnout dimension were selected for the present study. Responses were based on a 5-point Likert scale (1 = never; 5 = always). The average scale score was derived and used in the analyses. The scale properties obtained in the present study are reported in Table 1.

### 2.5. Data Analysis

Standard descriptive statistics were computed by using SPSS 26.0. We conducted Confirmatory Factor Analysis (CFA) by using Mplus version 8.8. to check for whether the four crucial study constructs (i.e., male partner workaholism, male partner burnout, female partner workaholism, and female partner burnout) could be discriminated empirically. The Actor-Partner Interdependence Model, including actor and partner effects, was tested through path analyses conducted with Mplus. First, we tested a baseline path analytic model including the 'actor effects' (i.e., the relationships male partner workaholism-male partner personal burnout and female partner workaholism-female partner personal burnout) as well as the correlations between the partner's level of workaholism and between the partner's level of residual personal burnout. After this, we tested the full APIM model including the 'partner effects' (i.e., the paths female partner workaholism-male partner personal burnout and male partner workaholism-female partner personal burnout). As standard practice, model fit was assessed in terms of the chi-square statistics and additional fit indices: comparative fit index (CFI), Tucker–Lewis index (TLI), root-mean-square error of approximation (RMSEA), and standardized root-mean-square residual (SRMR). Generally, TLI and CFI values greater than 0.90 and RMSEA and SRMR values lower than 0.08 are considered indications of acceptable fit [52]. However, the full APIM model was a saturated model (i.e., a model with zero degrees of freedom) reaching perfect fit, so fit statistics were used for the assessment of the baseline model only.

Finally, two further models were tested separately to analyze the moderating role of the presence of children on the workaholism–personal burnout relationships. These models were simplified versions of the full APIM model, specifically developed for testing the hypothesized interactions. With the first model, we looked at whether the interactions made by male partner workaholism and female partner workaholism with the number of children predicted male partner personal burnout. With the second model, we looked at whether the same two interactions predicted female partner personal burnout. These models were tested using the SPSS macro Process v2.16.3 [53] using centered versions of the predictors making up the interaction terms. The dataset used and the syntax of the analyses conducted are available upon request from the corresponding author.

## 3. Results

### 3.1. Preliminary Analyses

The values of internal consistency (Cronbach's alpha) were at least adequate for all the variables used in the APIM model (see Table 1). We then conducted Confirmatory Factor Analysis (CFA) to assess whether the four crucial study constructs (i.e., male partner workaholism/personal burnout and female partner workaholism/personal burnout) could be discriminated. In this analysis, the manifest indicators for workaholism were, in both cases (i.e., male partner workaholism and female partner workaholism), the four subscales of the MWS (see above), while the manifest indicators of personal burnout were, for both partners, the six items composing the personal burnout scale (see above). CFA results revealed that a 4-factor model (male partner workaholism, female partner workaholism, male partner personal burnout, and female partner personal burnout) fitted the data adequately: [$\chi^2(162) = 269.84$, $p < 0.001$; CFI = 0.91; TLI = 0.89; RMSEA = 0.070; SRMR = 0.085] and better than a 1-factor model [$\Delta\chi^2(6) = 471.40$, $p < 0.001$]. Although the values of the TLI and SRMR for the 4-factor model were slightly suboptimal, considering the obtained ratio $\chi^2/df$ (lower than 2) and the relatively small sample size (N = 138), we took the obtained results as adequate evidence that the four hypothesized constructs could be identified and discriminated sufficiently well. Before proceeding with the computation of correlations, two variables were recoded: Job position (1 = self-employed, managers or entrepreneur vs. 0 = employees) and educational level (1 = university level vs. 0 = up to high school). We then evaluated correlations between the main independent and dependent variables considered. As shown in Table 1, workaholism correlated positively with personal burnout in both male and female partners (respectively: r = 0.60, $p < 0.01$ for men; r = 0.45 $p < 0.01$ for women). This means that a higher level of workaholism is associated with a higher level of exhaustion and personal burnout symptoms for individuals of both sexes.

Moreover, workaholism and personal burnout, on the one hand, and high job position, on the other, were positively correlated in male partners (respectively, r = 0.34, $p < 0.01$; r = 0.20, $p < 0.05$). This means that greater job responsibilities are related to higher workaholism and personal burnout in men. Another interesting finding was the positive correlation between the level of education reached by both partners (r = 0.30, $p < 0.01$). Additionally, in both male and female partners, the level of education was significantly and negatively related to the number of children they had (r = −0.22, $p < 0.05$ and r = −0.29, $p < 0.01$ respectively).

### 3.2. Actor Partner Interdependence Model (APIM) Testing

We first tested a model including only actor effects, that is, the paths from male workaholism to male personal burnout and from female workaholism to female personal burnout. Model fit was the following: Chi-square (2) = 4.2, $p = 0.12$; CFI = 0.98; TLI = 0.94; RMSEA = 0.089; SRMR = 0.043. Given that a high RMSEA is a frequent occurrence with models characterized by a small number of degrees of freedom and low sample size such as ours [54], we considered the obtained model adequate. The results showed a significant workaholism-personal burnout relationship for both members of the couple. In fact, male workaholism positively related to the level of male personal burnout

(β = 0.601, *p* < 0.001). At the same time, females' workaholism predicted positively their personal burnout (β = 0.426, *p* < 0.001). In addition, the results indicated that male partner workaholism and female partner workaholism were significantly and positively correlated (r = 0.180; *p* = 0.034). The correlation between the residuals of male and female partner's personal burnout was also positive and significant (r = 0.171; *p* = 0.043). This indicated crossover processes at different levels in the investigated couples.

Then, to develop our model of interdependence between actors and partners, we evaluated the effects of actors, along with those of partners. Again (see Table 2 and Figure 2), the results confirmed that, for both male and female partners, the level of workaholism related positively with the level of personal burnout, which supported Hypothesis 1. We next examined the partner effects and results indicated that female partner workaholism was positively related to male partner personal burnout (β = 0.143; *p* = 0.038) over and above male partner workaholism. In contrast, male-partner workaholism was not significantly related to female partner personal burnout when controlling for female-partner workaholism (β = 0.005; *p* = 0.952). Therefore, the 'partner effect' was only confirmed for females. These results partially supported Hypothesis 2. The model explained 38% of the variance in male partner personal burnout and 20% of the variance in female partner personal burnout. The improvement in explained variance compared to the previously tested model, which did not include partner effects, was 2.1% in the case of male partner workaholism, while it was 1.8% in the case of female partner workaholism.

**Table 2.** Actor-Partner Interdependence Model.

| | Est. | S.E. | Est./S.E. | *p*-Value |
|---|---|---|---|---|
| Actor effects | | | | |
| Male partner whol → Male partner bout | 0.576 ** | 0.057 | 10.164 | 0.000 |
| Female partner whol → Female partner bout | 0.447 ** | 0.070 | 6.340 | 0.000 |
| Partner effects | | | | |
| Male partner whol → Female partner bout | 0.005 | 0.079 | 0.061 | 0.952 |
| Female partner whol → Male partner bout | 0.143 * | 0.069 | 2.071 | 0.038 |
| Male partner whol with Female partner whol | 0.182 * | 0.085 | 2.139 | 0.032 |
| Male partner bout with Female partner bout | 0.165 * | 0.083 | 1.976 | 0.048 |

Notes: Standardized results for APIM. Whol = workaholism; Bout = personal burnout. ** *p* < 0.01; * *p* < 0.05.

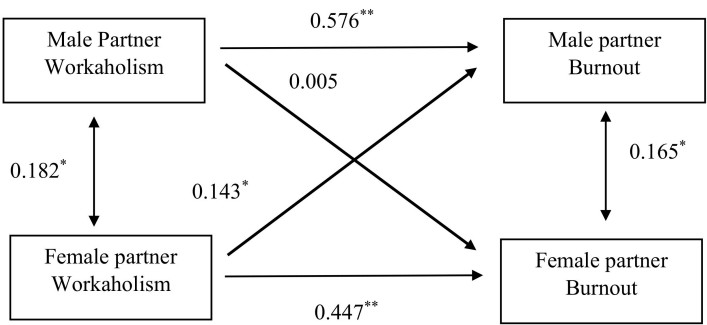

**Figure 2.** Graphical representation of the tested Actor-Partner Interdependence Model. ** *p* < 0.01; * *p* < 0.05. Note: Burnout in the figure refers to personal burnout.

### 3.3. Moderation

Moderation analysis targeting male partner personal burnout is reported in Table 3. With the inclusion of the interaction terms (female partner workaholism x number of children; male partner workaholism x number of children), the model explained 38% of the variance in male partner personal burnout. The results indicated that the number of children moderated (*b* = 0.157, *p* < 0.05) the relationship between the female partner's workaholism and the male partner's personal burnout (Table 3). Examination of the interaction graph showed that the relationship between female partner workaholism and

male partner personal burnout is stronger when the number of children in the couple is higher (Figure 3). Simple slope analysis indicated that when the number of children was higher (1 SD above the mean) there was a stronger relationship between female partner workaholism and male partner personal burnout, $b = 0.35$; $p < 0.01$, compared to when the number of children was lower (1 SD below the mean), $b = -0.07$; $p > 0.05$. Differently, the relationship between the male partner's workaholism and his own level of personal burnout was not moderated by the number of children ($b = 0.061$; $p > 0.05$) (Table 3).

**Table 3.** Model 1 Summary: Female partner workaholism, number of children, and male partner workaholism (covariate) on male partner personal burnout.

|  | $b$ | se | T | $p$ |
|---|---|---|---|---|
| Constant | 1.613 | 0.414 | 3.899 | <0.001 |
| Male partner whol | 0.445 | 0.116 | 3.849 | <0.001 |
| Female partner whol | −0.103 | 0.133 | −0.770 | 0.443 |
| N. of child. | −0.621 | 0.228 | −2.727 | 0.007 |
| Int_1 | 0.157 | 0.074 | 2.126 | 0.035 |
| Int_2 | 0.061 | 0.065 | 0.933 | 0.352 |

Notes: Whol = workaholism; N. of child. = number of children; Int_1 = first interaction term (female partner workaholism x number of children); Int_2 = second interaction term (male partner workaholism x number of children).

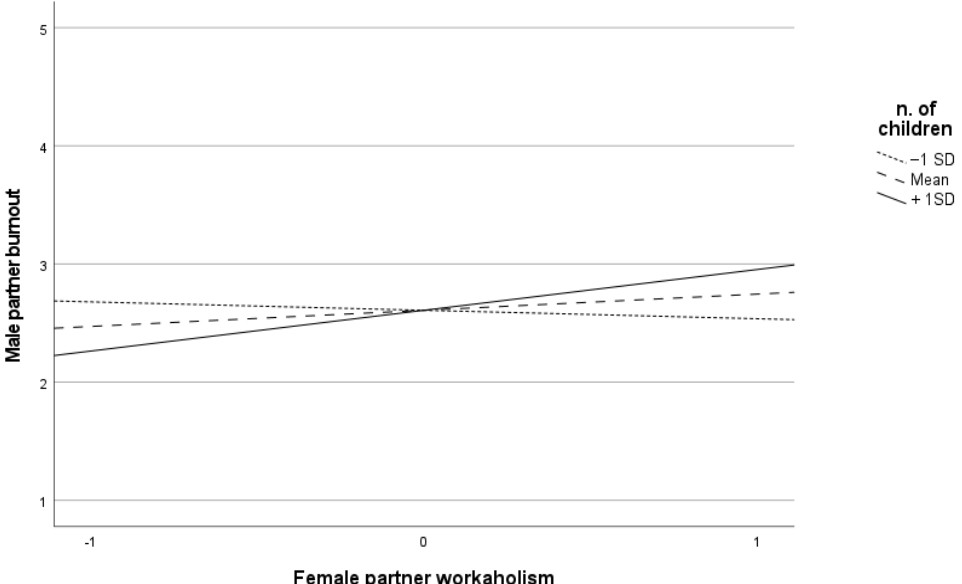

**Figure 3.** Plot of the relationship between female partner workaholism and male partner personal burnout moderated by the number of children.

The second model, used to predict female partner personal burnout, explained 25% of the variance with the inclusion of the interaction terms (male partner workaholism x number of children; female partner workaholism x number of children). The analysis (Table 4) revealed that the presence of children moderated the relationship between female partner workaholism and her own level of personal burnout ($b = -0.20$; $p < 0.05$), in the sense that this relationship declines when there is a higher number of children (see Figure 4). Simple slope analysis revealed that the relationship was significant when the number of children was low (1 SD below the mean), $b = 0.67$; $p < 0.01$, while it was not significant when the number of children was high (1 SD above the mean), $b = 0.23$; $p > 0.05$. Based on the moderation results, our last hypothesis (Hypothesis 5) was only partially confirmed.

**Table 4.** Model 2 Summary: Male partner' workaholism, number of children and female partner workaholism (covariate) on female partner personal burnout.

|  | *b* | **SE** | *t* | *p* |
|---|---|---|---|---|
| Constant | 1.044 | 0.482 | 2.165 | 0.032 |
| Male partner whol | −0.089 | 0.135 | −0.659 | 0.511 |
| Female partner whol | 0.763 | 0.155 | 4.907 | <0.001 |
| N. of child. | 0.053 | 0.265 | 2.002 | 0.047 |
| Int_1 | −0.202 | 0.086 | −2.354 | 0.020 |
| Int_2 | 0.034 | 0.076 | 0.442 | 0.659 |

Notes: Whol = workaholism; N. of child. = number of children; Int_1 = first interaction term (female partner workaholism x number of children); Int_2 = second interaction term (male partner workaholism x number of children).

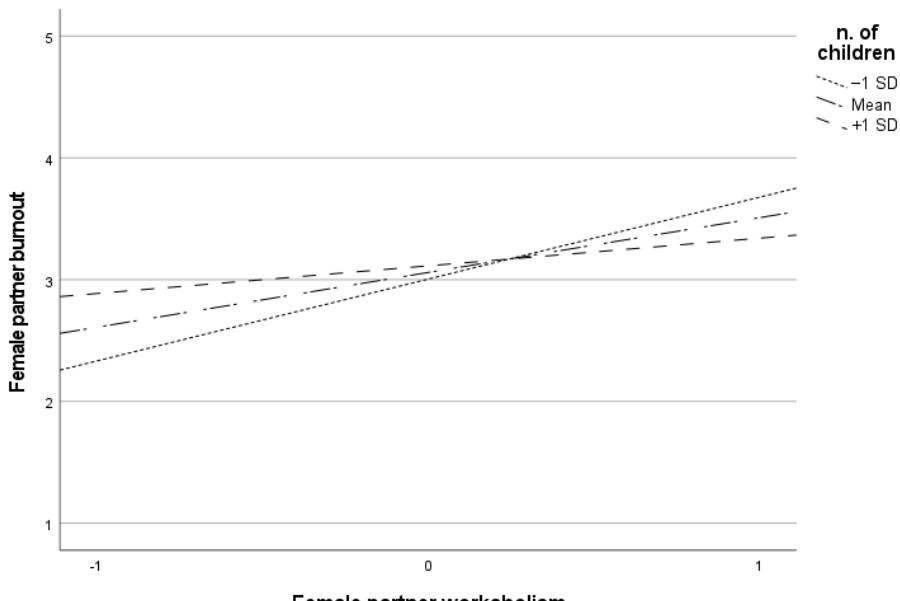

**Figure 4.** Plot of the relationship between female partner workaholism and female partner personal burnout moderated by the number of children.

## 4. Discussion

In line with our first hypothesis, we found that in both male and female partners, the reported level of workaholism was significantly related to symptoms of general burnout as operationalized in terms of emotional exhaustion. This result is not new in the literature, and it can be easily explained in terms of the heavy work investment exerted by individuals with higher workaholic tendencies compared to others. High work investment means time and effort investment [55], and effort-recovery theory [56] suggests that when an individual devotes a high effort at work and works for very long hours—which is quite typical for workaholics—affective strain is a logical and likely consequence. Since in workaholism a heavy investment at work is not a sporadic phenomenon but it is the rule, given that the behavior is driven by an internal and uncontrollable factor, this may lead to chronic emotional strain reactions such as those falling within the domain of personal burnout, which is a consequence of exposure to chronic stress [57]. This signals the unsustainability of the situations from a health and well-being point of view. Of importance is that the relationship between workaholism and personal burnout holds for both members of the couple, which suggests that workaholic partners may transfer within the family significant levels of emotional tension and distress, which is highly dysfunctional as far as the psychological family climate is concerned. Dysfunctions in the family of workaholics have already been discussed with a clinical approach [58], with findings indicating that spouses of workaholics experience greater marital estrangement and less positive affect than spouses

of non-workaholics, and anecdotal evidence from 'workaholic families' suggesting a high prevalence of broken marriages and brittle social relationships [59]. More recent empirical research confirmed these findings [39] and highlighted that the consequences for children may also be significant in terms of emotional and behavioral problems [45]. Personal burnout may be an important intervening mechanism for these outcomes, especially in dual-earner couples where both partners may experience workaholism and exhaustion symptoms. It is also of note that a couple in which both partners are high in workaholism may not be so infrequent, given our finding that workaholism in partners is positively and significantly related.

Our second hypothesis was that the level of workaholism reported by one member of the couple would be positively related to the level of personal burnout of the other member (i.e., partner effects). We found evidence only partially in line with this hypothesis since female partner workaholism was positively related to male partner personal burnout, whilst male partner workaholism was unrelated to female partner personal burnout. In other words, the negative crossover of workaholism was only identifiable in male partner personal burnout. This suggests that, for males, having a workaholic female partner makes a difference in terms of unwell-being, while for females having a workaholic male partner may be less important. Such disparity may perhaps be explained on the basis of traditional gender role expectations [60]. According to these, for males, it is 'normal' and accepted to invest heavily at work and act as the main breadwinner in the family, while for females, it is not so since they are expected to invest more in the family rather than at work. Such different gender role expectations are still very rooted in the Italian culture—i.e., the context of the present study—given that beliefs such as "for the man, more than for the woman, it is very important to be successful at work", "men are less suited to do housework" and "it is up to the man to provide for the family's financial needs", are very prevalent, with 58.8% of the population reporting to agree with at least one of them [61]. This means that a female partner with workaholic tendencies, who invests a great amount of time and effort at work, may destabilize the family to a greater extent compared to a male partner with the same tendencies, leading to more accentuated stress and exhaustion symptoms for the male partner. Intervening phenomena, which were not investigated in the present study, may also play a role here, such as that a male partner may be less prepared to deal with the additional work–family conflict issues [35] generated by a female partner with workaholic tendencies, who is mainly dedicated to work and less available for the family.

Moderation analysis revealed a pattern of results that were only partially in line with the formulated hypothesis that the presence and number of children would strengthen the investigated workaholism–personal burnout relationships. Indeed, the results revealed that the hypothesis was supported only with regard to the female partner workaholism-male partner personal burnout relationship. In this case, a higher number of children made a difference and accentuated the relationship. This is an original finding, indicating that in bigger families, having a female partner with workaholic tendencies may make life harder for the male partner, independently of the male partner's levels of workaholism, which were controlled for. The reason for this finding is that bigger families with more children may make the family system more complex and demanding, for example, for the management of children's schedules and activities; so in this case, a female partner with workaholic tendencies may lead to an additional commitment of the male partner. Following again the logic of role expectations theory [60], in bigger families, the lack of conformity to the expected role by the female partner with workaholic tendencies may be more detrimental to the group dynamics, amplifying the negative consequences for the other family members. On the contrary, the presence of children did not alter the relationship between male workaholism and personal burnout—either self-reported or reported by the female partner—which is again compatible with role expectations theory since caring for the family and children remains mainly a prerogative of the female partner. As a consequence, the male partner workaholism relationships with personal burnout are not significantly affected by the presence of children.

Interestingly, and contrary to our expectations, the presence of children acted as a protective factor for the relationship between female partner's workaholism and their own levels of personal burnout. More children meant a weaker relationship, which is exactly contrary to our hypothesis. This may suggest that, with children, the self-reported emotional costs of workaholism are reduced for female partners only; for example, because they may be able to access, more than their male counterparts, the social support aspects coming from the relationships with their children [62,63], leading to lower exhaustion symptoms. In other words, although workaholic women—compared to non-workaholic ones—may report a higher level of role conflict due to contradicting pressures determined by workaholism and gender role expectations, when there are children, they must perhaps reduce their involvement in work and can access unique social support resources that may be relatively beneficial for well-being. Additional non-measured variables may also come into play here, such as a stronger support system available to workaholic women with more children, which can help in reducing workaholism-related personal burnout.

*Implications and Limitations*

The present study suggests that workaholism should be prevented in organizations since it may intrude with negative consequences in the life of families in different ways, for example, by accentuating the workaholism level of the partners and by fueling personal burnout symptoms which may undermine an adequate social functioning of the family, as well as of the workaholic individual. To this end, different prevention stages may be considered [64]. From a primary prevention perspective, organizations should discourage overworking and promote disconnection from work and family-friendly policies; in other words, they should reinforce a balanced and sustainable working life. This is particularly important in the digital era, where one may work 24/7. Such prevention activities may create the conditions for workaholic predispositions [7,65] that some individuals may have, to remain silent. From a secondary prevention perspective, counseling services should be available that help individuals with workaholic tendencies to understand that their problematic work behavior may pose serious health risks, including leading to death from overwork-related disorders [66]. Following this, training on disconnection and recovery [67] may also be important as a way to promote healthier lifestyles from an individual perspective. Tertiary preventive interventions may still be based on counseling, focused psychological therapy, or dedicated self-help groups [65], with the aim of treating both workaholic consequences (e.g., personal burnout) or the psychological ingredients (e.g., obsessive-compulsive characteristics) that may fuel the phenomenon.

The present study has several strengths—such as the fact that it is based on multisource data and that it used the APIM model [32], which has rarely been applied in workaholism research, but it also has several limitations. First, it is a cross-sectional study that assessed the relationship between workaholism and burnout in men and women at a single time point. A longitudinal study with a defined time frame would complement our results and give insights into normal and reverse causation processes among the selected variables. In this way, it is also possible for future studies to assess the impact of the presence of children on the workaholism-personal burnout relationship over time. Second, the sample of our study was relatively small (137 couples) and consisted exclusively of Italian dual-earner couples. The generalizability of the results to other cultures and different types of couples (i.e., separated couples) should be explored by future investigations. In addition, the basic APIM model with four variables was used. Such a model could be expanded by future researchers to incorporate further variables that operate as intervening or moderating factors (e.g., work-family conflict, personality, and working conditions). The role of additional control variables could also be considered, such as, for couples, the duration of the relationship, which may be related to more effective coping strategies developed to manage stress [68]. This would allow us to reach more solid conclusions regarding mechanisms and boundary conditions explaining the relationships between workaholism and personal burnout in couples.

## 5. Conclusions

This study was an attempt to expand our understanding of the relationship between workaholism and burnout in dual-earner couples by means of the Actor-Partner Interdependence Model (APIM). It has shown that the levels of workaholism of partners are related and that for both partners, workaholism is related to burnout, suggesting a very detrimental impact of workaholism in the life of working couples and their families. Additionally, the female partner's level of workaholism crossed over and explains the male partner's personal burnout, which is additional evidence of the detrimental role of workaholism. The presence of children partially acted as a moderator of the investigated relationships, either accentuating or attenuating the potential impact of workaholism according to the partner's gender and the specific relationship examined in the APIM model. In both cases, however, children may be negatively impacted by workaholism and personal burnout, absorbing the negative influences coming from parents with workaholic tendencies.

**Author Contributions:** Conceptualization, C.B.; methodology, C.B.; formal analysis, E.R. and C.B.; investigation, C.B. and E.R.; data curation, C.B. and E.R.; writing—original draft preparation, E.R. and C.B.; writing—review and editing, P.A., S.T. and S.Z.; supervision, C.B. All authors have read and agreed to the published version of the manuscript.

**Funding:** This research received no external funding.

**Institutional Review Board Statement:** The study was conducted in accordance with the Declaration of Helsinki. Since there was no medical treatment or other procedures that could cause psychological or social discomfort to participants, who were all healthy adult subjects anonymously involved, additional ethical approval was not required.

**Informed Consent Statement:** Informed consent was obtained from all subjects involved in the study.

**Data Availability Statement:** Data and syntax are available from the last author of the study (C.B.).

**Conflicts of Interest:** The authors declare no conflict of interest.

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
