# Peer review of "The Relationship between Workaholism and Personal Burnout in Dual-Earner Couples: An Analysis Using the Actor-Partner Interdependence Model"

_sustainability, doi:10.3390/su151713009_

Round 1

Reviewer 1 Report

x) line 55: presenteeism should probably be considered among the negative consequences for the organisation (e.g., Girardi et al., 2015, doi: 10.4473/TPM22.4.5: ; Mazzetti et al., 2019, doi: 10.1002/ijop.12449).

x) line 184. It seems to me that there is a substantial overlap between gender and type of job (about 60/40). Is it possible that the results of the study are influenced to some extent by the type of job (e.g. men are more likely to be in a position that makes it easier to become a 'workaholic')?

x) line 243. I see descriptive statistics in Table 1, but no scale properties. I suggest reporting some data about factor structure and (lack of) common method bias, if the research is based on self-report data collected in a single occasion. However, I see at line 553 that the study "is based on multisource data", so I apologize for possible misunderstanding concerning this point.

x) line 258. So personal burnout, but no work-related burnout, was investigated in the study? If so, I would mention this in the title, abstract and discussion, since generally readers may think of burnout as "a syndrome conceptualized as resulting from chronic workplace stress that has not been successfully managed". Additionally, "Burn-out refers specifically to phenomena in the occupational context and should not be applied to describe experiences in other areas of life" (WHO, ICD-11; https://icd.who.int/browse11/l-m/en#/http://id.who.int/icd/entity/129180281), so readers could be surprised of not seeing job burnout in the study.

x) line 286. Why the interaction models were tested using linear regression? I think these two models could also have been tested using path analysis (possibly testing a single model), but please correct me if I am wrong.

x) line 343. To my understanding, another way of seeing the non significant p-value could be that the first model offers a preferable, more parsimonious explanation of observed correlations compared to the second model. Since the two degrees of freedom reflect the partner effects, fixing these path to zero in the second model does not deteriorate model fit. Stated differently, the partner effects does not seem to me to contribute to ameliorate model fit. Is it possible to see the in the text change of R2 from model 2 to model 1, attributable to partner effects?

x) line 351. Do the authors mean correlation between residuals in path analysis?

x) line 400. Given the nature of the "number of children", I think that simple slope could also be investigated at the specific number of children in the family (e.g., 0, 1, 2, ecc.).

x) Figure 3/Figure 4. Why -1SD/Mean/+1SD is repeated twice in the right panel? Furthermore, since the burnout score range from 1 to 5, why the y-axis ranged from 2 to 3.5 (depending on the figure)?

Author Response

Please see the attached response letter.

Reviewer 2 Report

Thank you this really very intereacting and up-to day study. The problem of balance of work and family, particularly in the world where both partners are working is very important for many western societies.

Although the presentation of the results and their discussion is quite clear, there are some comments to be considered:

1. The age range of the couples is too big. I can hardly imagine, how it is possible to compare couple aged 24-30 to a couple aged 65-70, especially, if the last one spent in marriage 30 to 50 years. It is well-established in aging psychology, that older adults' couples, especially those who had long marriage, had worked out their own coping-strategies for coping with stress, probably, with work stress as well. So I would suggest to control for age and marriage length, if such data are available, or at least to add to discussion section this point as limitation and future directions for investigation if you don't have this data now.

Nevertheless, I believe that even in the current presentation results are very significant and important for publication.

Some paragraphs, particularly in the background part, are "heavy". You can understand them, but it takes couple times to read it through.

Reviewer 3 Report

Research has found that

1) the level of workaholism of the partners is related,

2) for both partners, workaholism is associated with burnout,

The most important conclusions are:

1)the harmful impact of workaholism on the lives of working couples,

2) harmful effects of workaholism on the family.

The study is of great scientific importance.

Author Response

Dear Reviewer,

thanks a lot for appreciating our study.